# Few-Shot In-Context Imitation Learning
# via Implicit Graph Alignment

**Vitalis Vosylius and Edward Johns**
The Robot Learning Lab
Imperial College London
United Kingdom
`vitalis.vosylius19@imperial.ac.uk`

**Abstract:** Consider the following problem: given a few demonstrations of a task across a few different objects, how can a robot learn to perform that same task on new, previously unseen objects? This is challenging because the large variety of objects within a class makes it difficult to infer the task-relevant relationship between the new objects and the objects in the demonstrations. We address this by formulating imitation learning as a conditional alignment problem between graph representations of objects. Consequently, we show that this conditioning allows for in-context learning, where a robot can perform a task on a set of new objects immediately after the demonstrations, without any prior knowledge about the object class or any further training. In our experiments, we explore and validate our design choices, and we show that our method is highly effective for few-shot learning of several real-world, everyday tasks, whilst outperforming baselines. Videos are available on our project webpage at `https://www.robot-learning.uk/implicit-graph-alignment`.

**Keywords:** Few-Shot Imitation Learning, Graph Neural Networks

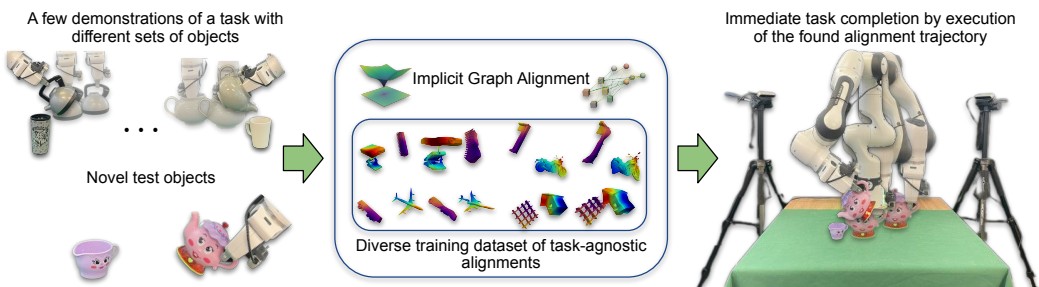

Figure 1: Given a few (e.g. three) demonstrations of a new task across multiple object instances (left), a robot can perform this same task with previously unseen objects (right), based on a graph alignment energy model trained on a diverse dataset of task-agnostic object alignments (middle).

## 1 Introduction

For a robot to perform a manipulation task involving two objects, it must infer task-relevant parts of these objects and align them throughout the trajectory in a task-specific way. For example, the spout of a teapot should be aligned with the opening of a mug when pouring tea. However, objects within a class typically have slightly different shapes, so inferring the parts which should be aligned for a given task is challenging due to the diversity of object shapes. This becomes even more challenging in an imitation learning setting, where this task-specific alignment should be inferred only from demonstrations, and ideally, without requiring any prior knowledge about the objects or their class.

We approach this challenge as a few-shot in-context imitation learning problem, and learn a conditional distribution of alignments that can produce task-relevant alignments of novel objects from just

7th Conference on Robot Learning (CoRL 2023), Atlanta, USA.

a few context examples, i.e. demonstrations. We propose to learn this distribution from point cloud observations using a novel heterogeneous graph-based energy model, whose graph structure with transformer-based attention is capable of representing intricate relations between different parts of the objects, and whose implicit energy-based learning is effective at capturing complex and multi-modal distributions. These design choices allow us to perform few-shot imitation learning on any arbitrary objects and over any arbitrary number of demonstrations, using just a single trained model.

But where can we obtain a large-scale dataset of task-relevant alignments between objects needed to train such a model? Our insight here is that simply by deforming pairs of arbitrary ShapeNet [1] objects in simulation, using an object augmentation function trained for a single object category of chairs [2], we can create sensible new instances of objects of an arbitrary category and align them consistently, providing us with the data required to train the aforementioned model. By training on objects from diverse categories and multiple alignments between them, we are able to predict task-relevant alignments for novel object pairs, all from the same trained model.

We evaluated our method both with simulation and real-world experiments and analysed its ability to align novel objects in a few-shot imitation learning setting, where each demonstration consists of one or more waypoints in a trajectory. Our real-world results show that for everyday tasks, such as pouring from a teapot into a mug, sweeping with a brush, and hanging a cap onto a peg, we achieve 80% task success rate under the following conditions: (1) only four demonstrations are provided, (2) the test objects are novel and unseen during the demonstrations, and (3) we assume no prior knowledge about the objects or class of objects. These results, when compared to the baselines, validate our method as an effective few-shot imitation in-context learning framework, capable of efficiently learning everyday robotic tasks from just a few demonstrations.

## 2 Related Work

The field of imitation learning has seen various approaches that aim to solve robotic tasks based on a single demonstration [3, 4, 5, 6]. However, these methods are typically limited to scenarios where the objects used in the demonstrations and during deployment are either the same or sufficiently similar. This lack of flexibility prevents them from being applied to category-level manipulation tasks. Other methods tackle this limitation by leveraging category-level keypoints, which can be annotated [7, 8], predicted [9], or obtained through self-supervised correspondence methods [10, 11, 12]. However, selecting such keypoints in a task-agnostic manner is challenging, thereby restricting the applicability of these approaches. Transporter Nets [13], utilises learned feature maps and template matching to identify pick-and-place poses, but is restricted to pick-and-place tasks and struggles to predict arbitrary 6DoF poses. We also train a function to predict the quality of an object alignment, but we do so in an unrestricted $\mathbb{SE}(3)$ space, enabling us to handle much more complex and arbitrary tasks defined by the demonstrations. Graph neural networks [14, 15, 16, 17] and implicit learning of functions [18, 19, 20, 21] have recently gained significant attention for robot manipulation, and we study a novel formulation of these ideas for few-shot imitation learning.

The most closely related work to ours comes from Relational Neural Descriptor Fields (R-NDF) [22, 23] and TAX-Pose [24], both of which can generalise to novel instances within the same object category. [22] achieves object alignment by matching descriptors extracted from a pre-trained occupancy network. However, it requires per-object category training and the annotation of task-relevant object parts. Furthermore, [22] averages descriptors from multiple demonstrations, which can pose challenges when dealing with significantly different objects or multimodal demonstrations. [24] employs cross-object attention for estimating dense cross-object correspondences. However, it requires separate network training for each new alignment between the objects, limiting its applicability. In contrast to previously described methods, our approach learns a general conditional distribution of alignments using an energy-based model, which can be applied to objects of unseen categories and arbitrary alignments without the need for additional training beyond the initial demonstrations.

## 3 Method

### 3.1 Problem Formulation & Overview

**Problem Setting.** We consider an imitation learning setting involving two objects, namely a *grasped object* (or a robot gripper) and a *target object*, between which the interaction should occur. We do

not assume any prior knowledge about these objects, and denote their observations as $O_A$, and $O_B$. The alignment, or relative spatial configuration between these objects, is denoted as a general representation $\mathcal{A}(O_A, O_B)$, which should be independent of the global configurations of the objects, and focusses only on their relative alignment. As we will explain later, we use segmented point clouds as observations $O_A$ and $O_B$ and devise a heterogeneous graph to represent $\mathcal{A}(O_A, O_B)$.

**Objective.** Given an arbitrary task involving any two objects $O_A$ and $O_B$, our goal is to infer a trajectory of alignments between them that would complete the task. We denote this trajectory as a sequence of alignments $\mathcal{A}_{test}^{1:T}(O_A, O_B)$, where $T$ is the number of time steps in a trajectory. In general, to accomplish this, we would require knowledge of the task-conditional distribution $p(\mathcal{A}_{test}^{1:T}|task)$, but defining this analytically is infeasible for all possible task and object combinations and from partial point cloud observations. Instead, we propose to approximate it using a few samples from this distribution $\{\mathcal{A}_{demo}^{1:T}(O_A^i, O_B^i)\}_i^N$, by learning a conditional distribution $p_\theta(\mathcal{A}_{test}^{1:T}|\{\mathcal{A}_{demo}^{1:T}\}_i^N)$, parameterised by $\theta$. Here, $N$ denotes the number of demonstrations.

**Deployment.** When presented with novel objects $O_A$ and $O_B$, our objective is to perform a task by sampling from the learned conditional distribution, eliminating the need for task-specific training or fine-tuning. We sample from $p_\theta(\mathcal{A}_{test}(\mathcal{T}^t \times O_A, O_B)|\{\mathcal{A}_{demo}^t\}_i^N))$ by finding a sequence of transformations $\mathcal{T}^{1:T}$ which when applied to object $A$, yield alignments within the distribution. Assuming rigid grasps, we can directly apply this sequence of transformations to the robot's end-effector, causing it to move the grasped object along the task-relevant trajectory. In this work, we treat individual time steps in a trajectory independently, hence, going forward, we will drop the superscript $t$, and refer to a set of $N$ demonstrations as $\mathcal{A}_{demo}$ for conciseness.

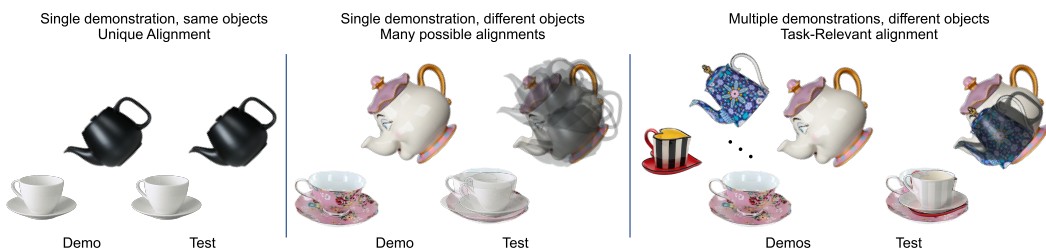

Figure 2: Illustration motivating the need for multiple demonstrations across different sets of objects, in order to generalise to unseen objects when inferring task-relevant alignments.

**Multiple Demonstrations Requirement.** When the objects during test time are the same as during the demonstrations, sampling from the learned distribution would yield alignments identical to the demonstrations (see Figure 2 left). However, if they differ from those in the demonstrations, the task-relevant parts should be aligned consistently. Inferring relevant object parts for an arbitrary task is infeasible from a single demonstration when objects used are different (see Figure 2 middle). Therefore, we use multiple demonstrations with different objects from the same categories and make an assumption that across the demonstrations, the relative alignment of the task-relevant parts remains consistent and can be inferred (see Figure 2 right). With an increasing number of demonstrations and a broader range of object shapes, the conditional distribution should converge towards a Dirac delta function $\delta(\cdot)$. In cases where the demonstrations are multimodal or inconsistent, the distribution becomes multimodal, representing different possible ways of completing the task.

### 3.2 Generalisation Through Shape Augmentation

To learn a conditional distribution $p_\theta(\mathcal{A}_{test}|\mathcal{A}_{demo})$ which generalises to arbitrary alignments between different instances of objects from the same category, we need a diverse dataset containing multiple objects with different geometries aligned in a semantically consistent way. Obtaining such a dataset would require a huge effort in annotation or solving the underlying semantic object alignment problem in the first place. Instead, we propose to utilise correspondence-preserving shape augmentation and treat each deformation of the object as a new instance. The ability to generate new instances of a specific category of objects and having privileged information regarding the ground truth correspondences allows us to create an arbitrary number of alignments between the objects with specific parts consistently aligned (see Figure 3). Although these deformations might not

represent realistic objects one might encounter in the real world, we hope that we can encapsulate the true underlying distribution of objects within the convex hull of the produced deformations.

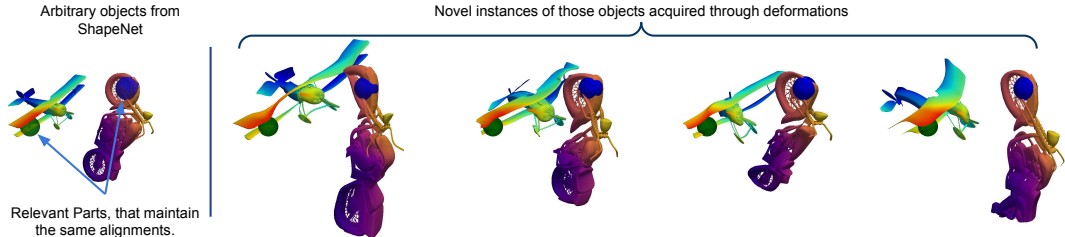

Arbitrary objects from ShapeNet

Novel instances of those objects acquired through deformations

Relevant Parts, that maintain the same alignments.

Figure 3: Two arbitrary objects and their deformed variations aligned based on specific parts. Colour maps indicate dense correspondences, green and blue spheres highlight the parts used for alignment.

In practice, we create a diverse dataset of alignments, as follows. 1) *Random Object Selection*: We randomly select two objects from the ShapeNet dataset — representing $O_A$ and $O_B$, and apply random $\mathbb{SE}(3)$ transformations to them, creating an arbitrary alignment between them. 2) *Deformation Generation*: To create variations of objects, we employ a correspondence-preserving implicit generative model called DIF-Net [2]. By deforming the objects, we obtain different instances of the same category, each with its own unique shape characteristics. 3) *Alignment Based on Correspondences*: For each original object, we randomly sample a point on its surface. Using the known correspondences between the original and deformed objects, we align the deformed objects based on the local neighbourhood of points around the sampled point. This alignment ensures that certain parts of the objects maintain the same relative pose consistently. 4) *Rendering Point Clouds*: Finally, we render the point clouds of the aligned objects using simulated depth cameras, resulting in a sample of consistently aligned object pairs. The resulting dataset $\mathcal{D} = \{\{\mathcal{P}_A^i, \mathcal{P}_B^i\}_i^N\}_j^M$ contains $M$ samples, where each sample consists of $N$ aligned object pairs. We utilise these samples by using $N-1$ alignments as context and the left-out alignment as the target alignment for those object.

## 3.3 Alignment Representation

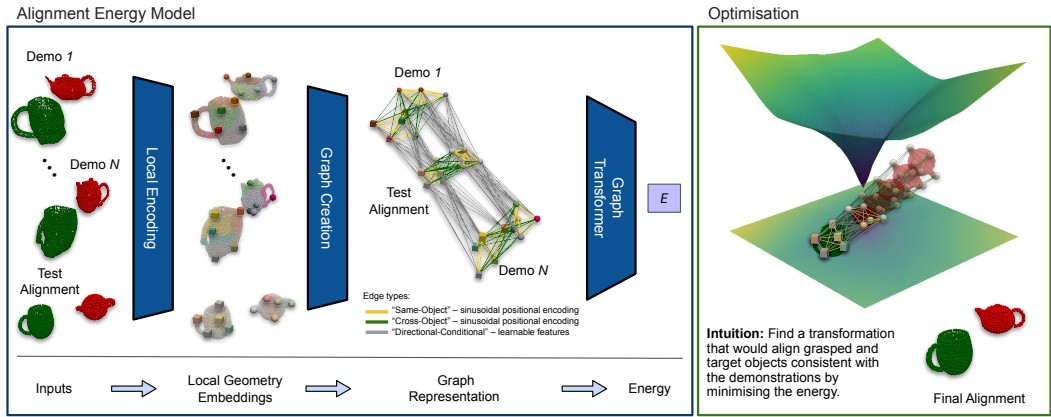

Figure 4: (Left) High-level structure of the model used to learn the alignment distribution. (Right) The learnt energy landscape in 2D, and the graph alignment during different optimisation steps.

As hinted previously, we use point clouds $\mathcal{P}_A^i$ and $\mathcal{P}_B^i$ as the initial representation of the objects. To learn the proposed distribution $p_\theta(\mathcal{A}_{test}|\mathcal{A}_{demo})$, we need to reason about the geometries of different parts of the objects and their relative alignments. Directly using dense point clouds would make this problem extremely computationally expensive. Therefore, we design a representation for $\mathcal{A}(O_A, O_B)$, that encodes different parts of the objects and the relative poses between them.

First, we sample $K$ points from each point cloud using Furthest Point Sampling Algorithm and encode local geometry around them using the Set Abstraction (SA) layer [25], to obtain a set of feature vectors $\mathcal{F}$ and their positions $p$ describing the whole object $\{\mathcal{F}^i, p^i\}_i^K$ (see Figure 4). We utilise

Vector Neurons [26] which, together with the local nature of the SA layer, provide us with $\mathbb{SE}(3)$-Equivariant embeddings $\mathcal{F}$. To ensure that these embeddings describe the local geometry around the sampled point, we pre-train this network as an implicit occupancy network [27], reconstructing the encoded object. Note that each local embedding is used to reconstruct only a part of the object, reducing the complexity of the problem and allowing it to generalise more easily. We provide a full description of the occupancy network used in Appendix B. To encode the relative alignment between the two objects based on the sets of feature and position pairs, we then construct a heterogeneous graph representation. This graph captures the relationships between different parts of the objects and fully describes their geometry. Each edge $e_{ij}$ is assigned a feature vector representing the relative position between the nodes. To model high-frequency changes in the relative position between the nodes, we utilise positional encoding [28] to express edge features.

## 3.4 Learning the Conditional Alignment Distribution

We use our devised alignment representation $\mathcal{A}$ and learn the proposed conditional distribution $p_\theta(\mathcal{A}_{test}|\mathcal{A}_{demo})$ by employing a novel heterogeneous graph-based transformer energy model denoted as $E_\theta(\mathcal{A}_{test}, \mathcal{A}_{demo})$. Intuitively, the energy-based model should compare $\mathcal{A}_{test}$ with $\mathcal{A}_{demo}$, and predict if alignments are consistent (low energy) or not (high energy). Then, by minimising the predicted energy with respect to $\mathcal{A}_{test}$, we can find a desired alignment of test objects.

Firstly, we combine the graphs describing $\mathcal{A}_{test}$ and $\mathcal{A}_{demo}$ by connecting nodes from demonstrated and test alignments with directional edges, equipped with learnable embeddings (see Figure 4). These connections effectively propagate information about the relative alignment of the objects during the demonstrations to the test observation. This enables the graph to capture the alignment patterns observed in the context and makes it suitable for making predictions about whether the test observation is consistent with the demonstrations. To propagate the information through the graph in a structured manner, we utilise graph transformer convolutions [29], which, for a heterogenous graph, can be viewed as a collection of cross-attention modules. We then learn $p_\theta(\mathcal{A}_{test}|\mathcal{A}_{demo})$ by employing an InfoNCE-style [30] loss function, and minimise the negative log-likelihood of:

$$\hat{p}_\theta\left(\mathcal{A}_{test}|\mathcal{A}_{demo}, \{\hat{\mathcal{A}}_{test}^j\}_j^{N_{neg}}\right) = \frac{\exp(-E_\theta(\mathcal{A}_{test}, \mathcal{A}_{demo}))}{\exp(-E_\theta(\mathcal{A}_{test}, \mathcal{A}_{demo})) + \sum_j^{N_{neg}} \exp(-E_\theta(\hat{\mathcal{A}}_{test}^j, \mathcal{A}_{demo}))}$$
(1)

Here $N_{neg}$ is a number of counter-examples, used to approximate the, otherwise intractable, normalisation constant. We create these counter-examples by applying $\mathbb{SE}(3)$ transformations to nodes in the graph describing $O_A$, transforming both their position $p_A$ and feature vectors $\mathcal{F}_A$. Note that we are able to do this due to the $\mathbb{SE}(3)$-equivariant property of our point cloud encodings. We obtain the transformations using a combination of random and Langevin Dynamics [31, 32] sampling. At inference, we sample from the learnt distribution by minimising the energy using gradient-based optimisation [33, 31] to find an $\mathbb{SE}(3)$ transformation $\mathcal{T}$ which, when applied to $O_A$, would result in an alignment between the objects which is consistent with the demonstrations (see Figure 4 right).

$$\mathcal{T} = \underset{\mathcal{T} \in SE(3)}{\operatorname{argmin}} E_\theta(\mathcal{A}_{test}(\mathcal{T} \times O_A, O_B), \mathcal{A}_{demo})$$
(2)

Equation 2 can be understood as follows: Adjust the pose of the object $A$ iteratively until the energy is minimised, indicating that the alignment between the objects $A$ and $B$ becomes consistent with the demonstrated alignments. In practice, we use separate networks for finding orientation and translation alignments, optimising them sequentially. In Appendix C, we provide full details regarding the training of the proposed energy model, and optimisation on the $\mathbb{SE}(3)$ manifold. Using a graph neural network with transformer-based attention allows us to capture complex relational information between different parts of two objects, and dynamically handle an increasing number of demonstrations during inference. The energy-based approach allows us to learn complex and multi-modal distributions. It can capture the inherent variability and uncertainty in the alignment process, allowing for different ways of aligning objects. This flexibility is crucial when multiple valid alignments exist, e.g. when dealing with symmetrical objects or multi-modal demonstrations.

# 4  Experiments

To thoroughly assess the effectiveness of our method in learning a general alignment distribution and its practicality in real-world robotic tasks, we conducted experiments in two distinct settings: (1) a simulation setting, where we have access to ground truth information, and (2) a real-world setting, where we evaluate the performance of the complete robotic pipeline on everyday robotic tasks. A single model trained on $500K$ samples (5 consistent alignments each) is used for all the experiments, evaluating it by providing new demonstrations as context. Videos are available on our project webpage at https://www.robot-learning.uk/implicit-graph-alignment.

## 4.1  Exploring Generalisation Modes

Our first set of simulated experiments aims to answer the following question: *Is our model capable of generalising and identifying task-relevant alignments when presented with both novel objects from familiar categories and entirely novel object categories?*

**Experimental Procedure.**  We constructed five distinct evaluation datasets, each introducing an additional mode of generalisation requirement. The generation procedure follows the methodology outlined in Section 3.2, and only includes objects from non-symmetric categories in order to evaluate the orientation accuracy of found alignments. The five evaluation datasets encompass the following samples: 1) *Seen Alignments*: This dataset comprises objects and their alignments that were seen during training. 2) *Unseen Alignments*: In this dataset, objects that were seen during training are aligned in novel ways, introducing unfamiliar alignment configurations. 3) *Unseen Object Instances*: This dataset contains unseen instances of object categories that were encountered during training. 4) *Unseen Object Categorie*s: Here, objects are from categories that were unseen during training. 5) *Multi-Modal Demonstrations*: This dataset focuses on multimodal demonstrations of objects from unseen categories. This means that the demonstrations exhibit multiple modes, indicating different possible object alignments. *Unseen*, here refers to objects, alignments or categories, that have not been used during training. Every described dataset contains 1000 unique samples, each sample with 5 consistent object alignments, 4 of which are used as the context, and 1 as a ground truth label.

**Evaluation Metrics.**  Having access to the ground truth alignment of the objects, we report the translation and rotation errors of the predicted alignment, when starting point clouds are initialised in random $\mathbb{SE}(3)$ poses. In the case of multi-modal demonstrations, we report the error to the closest ground truth alignment. To put the later presented error metrics in perspective, object sizes in the previously described datasets range from 8 cm to 35 cm.

**Baselines.**  In our evaluation, we compare our method of *Implicit Graph Alignment (IGA)* against several baselines and its own variants to demonstrate its effectiveness. 1) *ICP*, where the target and grasped object point clouds are aligned to the closest match amongst the available demonstrations. It serves to validate the need to learn the alignment distribution. 2) *TAX-Pose-DC*, is a variant of TAX-Pose-GC [24], where instead of one-hot encoding, we use an average embedding of the demonstrations as a conditional variable. For a fair comparison, this is our attempt to extend a cross-object correspondence approach to be conditional on demonstrations (the original TAX-Pose-GC conditional on a one-hot encoding of a desired placement). 3) *R-NDF* is Relational Neural Descriptor Fields [22], which aims to validate our choice of using a heterogeneous graph representation for the alignment of objects, instead of matching local descriptors of two objects as in R-NDF. 4) *Ours(DirReg)*, is a variant of our method which, using our graph representation, directly regresses the transformation $\mathcal{T}$ instead of learning an energy function. 5) *Ours(DFM)* is another variant of our method, which predicts $\mathbb{SE}(3)$ distance to the ground truth alignment and learns a distance field instead of an energy function. For a fair comparison, our method and all baselines were trained on the same amount of data from a diverse range of object categories.

| | (1) Seen Alignments | | (2) Unseen Alignments | | (3) Unseen Object Instances | | (4) Unseen Object Categories | | (5) Multi-Modal Demonstrations | |
|---|---|---|---|---|---|---|---|---|---|---|
| | Trans (cm) | Rot (deg) | Trans (cm) | Rot (deg) | Trans (cm) | Rot (deg) | Trans (cm) | Rot (deg) | Trans (cm) | Rot (deg) |
| ICP | 15.8 ± 7.7 | 77.9 ± 27.1 | 15.3 ± 10.3 | 78.5 ± 28.1 | 14.9 ± 8.7 | 69.4 ± 29.0 | 16.5 ± 8.9 | 99.7 ± 34.9 | 15.0 ± 10.1 | 83.3 ± 26.4 |
| R-NDF | 8.5 ± 3.2 | 43.1 ± 32.3 | 8.1 ± 1.7 | 37.9 ± 21.8 | 9.2 ± 6.3 | 46.5 ± 23.1 | 9.8 ± 5.4 | 52.1 ± 29.2 | 13.6 ± 7.6 | 96.4 ± 45.5 |
| Tax-Pose-DC | 21.8 ± 9.9 | 105.6 ± 39.2 | 19.6 ± 10.4 | 117.2 ± 36.8 | 24.3 ± 9.8 | 105.4 ± 36.7 | 21.3 ± 8.9 | 126.3 ± 38.1 | 23.9 ± 9.4 | 123.3 ± 36.7 |
| Ours (DirReg) | 4.1 ± 2.4 | 27.7 ± 6.4 | 4.9 ± 3.6 | 27.5 ± 9.6 | 5.2 ± 3.5 | 31.6 ± 11.2 | 6.7 ± 5.0 | 31.1 ± 10.7 | 13.1 ± 6.3 | 76.8 ± 26.4 |
| Ours (DFM) | 3.2 ± 1.2 | 89.6 ± 42.8 | 3.7 ± 1.6 | 106.1 ± 57.1 | 3.7 ± 1.5 | 103.4 ± 56.8 | 4.4 ± 2.0 | 115.3 ± 61.8 | 10.4 ± 5.1 | 114.9 ± 58.7 |
| Ours (IGA) | **2.1 ± 1.3** | **9.7 ± 6.4** | **2.3 ± 1.5** | **12.1 ± 5.6** | **2.3 ± 1.4** | **11.7 ± 6.1** | **3.0 ± 1.6** | **14.7 ± 5.3** | **3.9 ± 2.3** | **11.2 ± 6.7** |

Table 1: Translation and rotation errors (means and standard deviations) of alignments for different modes of generalisation. Values of the best-performing method are presented in bold.

**Results.** Results shown in Table 1 reveal that using non-learning-based methods, such as *ICP*, and extending the cross-object correspondence approach to be conditional, yield unsatisfactory performance. The sub-optimal performance of *R-NDF* can be attributed to the high diversity within object categories and instances, as well as the arbitrary alignments between them (as it primarily focuses on finding alignments between objects in close proximity). Other variants of our approach show promising results in terms of translation, but struggle to accurately determine the orientation. Our proposed method, using an implicit energy-based model, excels in both translation and orientation, and across different modes of generalisation.

## 4.2 Exploring Demonstration Diversity

Our second set of simulated experiments aims to answer the following question: *Which is more important: the number of demonstrations, or the diversity of demonstrations?*

**Experimental Procedure.** We explore this question by varying both the diversity and the number of demonstrations provided as context. We create 3 different evaluation datasets of demonstrations, each with increasing diversity of objects. We vary the diversity of the objects by increasing the random scaling range, as well as the scale of the latent vector $\alpha$, which controls the amount of deformations applied by the DIF-Net model [2]. For visualisation of the diversity of objects in the created evaluation datasets see Appendix D.4.

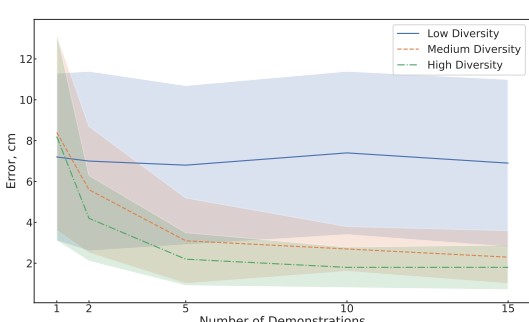

Figure 5: Translation error based on the number of demonstrations for 3 different sets of diversities.

**Results.** Translation errors are shown in Figure 5 (orientation errors follow a similar trend and can be seen in Appendix D.4). Note that a single model is used for all the evaluations. Analysis of Figure 5 reveals that the performance of our method does not improve when the number of demonstrations increases, unless the diversity among those demonstrations also increases. However, an increase in diversity can substantially enhance the performance. Notably, the performance of our method remains relatively unchanged when provided with either 10 or 15 demonstrations, suggesting that the remaining errors are not attributable to insufficient information in the demonstrations. Instead, they likely stem from inherent errors of the learnt model or ambiguities present in the generated datasets.

## 4.3 Real-World Experiments

Through our real-world experiments, we aim to answer the following question: *Can a robot learn real-world everyday tasks from just a small number of human demonstrations?*

**Real-World Setup.** Our experiments are performed using the Franka Emika robot, along with three RealSense D415 depth cameras. To obtain point clouds of the objects, the cameras first capture the point cloud of the target object with the robot not obstructing their view. Then, the robot moves its gripper to a position where the grasped object is visible to both cameras, after which the cameras capture a point cloud of the grasped object.

**Tasks.** We evaluate our approach on six different tasks, which can be seen in Figure 6. 1) *Grasping*. The goal is to grasp different pans by the handle. 2) *Stacking*. The goal is to stack two bowls. 3) *Sweeping*. The goal is to sweep marbles into a dustpan with a brush. 4) *Hanging*. The goal is to hang a cap onto a deformable stand. 5) *Inserting*. The goal is to insert a bottle into a shoe. 6) *Pouring*. The goal is to pour a marble into a mug. We provide success criteria in Appendix D.1.

**Experimental Procedure.** Trajectories for *Grasping* and *Stacking* tasks are defined by a single waypoint at the grasping or placing locations and executed with a scripted controller, consistent with *R-NDF* [22] and *TAX-Pose* [24]. For these tasks, we provide 3 demonstrations. Trajectories for other tasks are defined by 3 waypoints, which need to be reached sequentially. For these tasks, we provide 4 demonstrations each. Each demonstration is a unique combination of different grasped

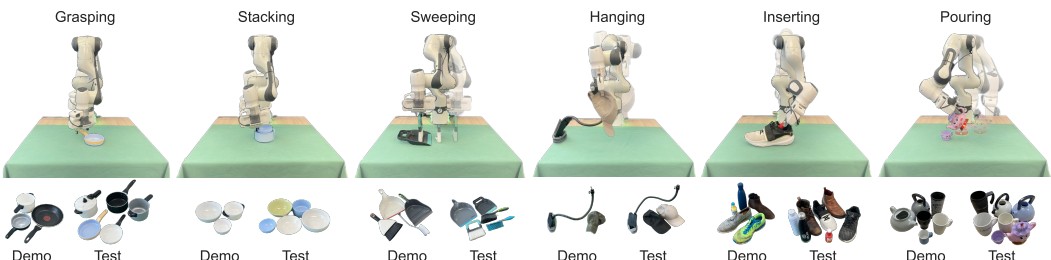

Figure 6: Real-world tasks used to evaluate our method, and the corresponding objects used for providing demonstrations and testing.

and target objects. Notably, the *Hanging* task involves a deformable stand and deformable cap, both of which are randomly deformed before each demonstration and test. We evaluate each task on 5 different combinations / deformations of grasped and target test objects, repeating the evaluation 3 times by starting the objects in different configurations, totalling 15 evaluations per task per method. No objects used to provide the demonstrations are used during the evaluation.

**Results.** As we can see from Table 2, *ICP* and *R-NDF* baselines can be successfully used to complete *Grasping* or *Stacking* tasks but struggle with more challenging ones, e.g. Hanging or Pouring. This can be attributed to the relatively high task tolerances and limited variation in the object geome-

|  | Grasp | Stack | Sweep | Hang | Insert | Pour | Avg. |
|---|---|---|---|---|---|---|---|
| ICP | 9 / 15 | 14 / 15 | 8 / 15 | 0 / 15 | 7 / 15 | 2 / 15 | 44 ± 31% |
| R-NDF | 7 / 15 | 14 / 15 | 4 / 15 | 0 / 15 | 4 / 15 | 3 / 15 | 36 ± 29% |
| IGA | 14/ 15 | 15 / 15 | 8 / 15 | 12 / 15 | 11 / 15 | 12 / 15 | 80 ± 15% |

Table 2: Success rates of our method and two best performing baselines in a real-world setting.

tries between demonstrations and inference for these tasks. Our method, on the other hand, can consistently complete all the evaluated tasks from just a few demonstrations, with the notable exception of a Sweeping task. We hypothesise that this is due to major geometry differences between demonstration and test objects, that were not covered by the shape augmentation model.

## 5    Discussion

**Limitations.** We now share the main limitations of our method, in order of importance. 1) As with many similar works, we assume that segmented point clouds are available of sufficient observability to capture task-relevant object parts, and thus we require three depth cameras. However, our experiments showed that with real-world hardware and noisy observations, our method is still highly effective for everyday tasks. 2) Our current implementation of gradient-based sampling is not yet suitable for reactive closed-loop control on today's computing hardware, as inference can be time-consuming (up to 15 seconds). 3) We model trajectories as sparse waypoints, which may not be suitable for tasks requiring more complex dynamics. However, in future work, our method could be extended to track a much denser trajectory, including velocities. 4) We assume that objects remain rigidly grasped during a trajectory. Therefore, whilst we have shown that our method works with deformable objects (the *Hanging* task), we have not yet extended this framework for objects which may deform during a task. 5) Our current formulation models a task as the interaction of 2 objects, which might not always be the case. However, we could extend our method to an arbitrary number of objects, if $O_B$ is used to represent an observation of the entire environment. 6) Training of our model relies on data generated using a heuristic method, which may introduce inaccuracies in the training samples and limit the model's precision.

**Conclusions.** We have proposed a method for few-shot imitation learning, which is able to perform real-world everyday tasks on previously unseen objects, from three or four demonstrations of that task on similar objects. We achieved this by designing a novel graph alignment strategy based on an energy-based model and in-context learning, which was empirically shown to outperform alternative methods. Our experiments and videos highlight that this is also a practical method which enables a robot to perform a task immediately after the demonstrations, without requiring any further data collection or any prior knowledge about the object or class, using just a single trained network.

**Acknowledgments**

The authors thank Norman Di Palo, Kamil Dreczkowski, Ivan Kapelyukh, Pietro Vitiello, Yifei Ren, Teyun Kwon, and Georgios Papagiannis for their valuable contributions through insightful discussions. Additionally, special thanks to Georgios Papagiannis for developing the robot controller used in our real-world experiments.

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

# Appendix

## A  Framework Overview

To learn our proposed conditional distribution of alignments as an energy-based model, we utilise various techniques to create the diverse dataset needed, pre-train an implicit occupancy network to capture local geometries of objects, and create a heterogeneous graph representation of object alignments. We discuss these techniques and their implementation details in the following sections, while the overview can be seen in Figure 7.

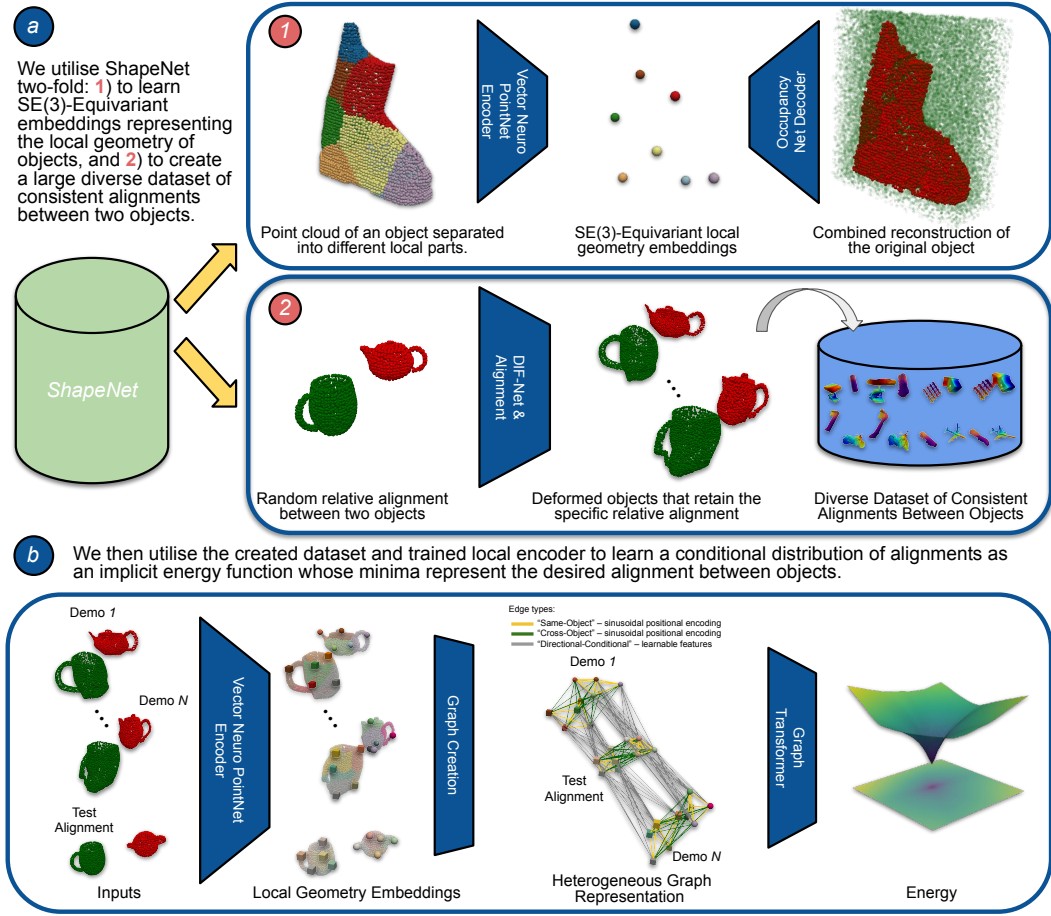

Figure 7: An overview of the techniques used in the design of our proposed framework.

## B  Local Encoder Network

The representation of object alignment $\mathcal{A}(O_A, O_B)$ as a heterogeneous graph is a crucial step in effectively capturing the relationship between the two objects. To achieve this, we begin by encoding the segmented point clouds of the objects as sets of feature and position pairs. The underlying assumption is that each feature vector can effectively represent the local geometry of a specific part of the object. By treating these feature vectors as nodes in a graph and connecting them with edges that represent their relative positions, we create a graph representation that enables the network to focus on the specific parts of the objects. By adopting this graph-based approach, we are able to shift the network's attention towards local information and individual parts of the objects, rather than relying solely on the global geometry of the entire objects. This localised representation facilitates more precise and targeted reasoning about the alignment between the objects, leading to improved performance in capturing the complex relationships and relative positions between the object parts.

To construct the local features, we first utilise the Furthest Point Sampling (FPS) algorithm to sample $K$ points on the surface of the point cloud (8 in our implementation). These sampled points serve as the centre positions $p_i$ for the subsequent calculation of local embeddings. Next, we group all the points in the original point cloud according to their closest centroid and re-centre them around their respective centroids. This grouping process results in $K$ different point clouds, each representing a distinct part of the object. To encode these local point clouds, we employ a shared PointNet model. This model takes each local point cloud as input and generates a feature vector $\mathcal{F}_i$ that describes the local neighbourhood around each centroid. Our PointNet model consists of an eight-layer MLP (Multi-Layer Perceptron) with skip connections, serving as the backbone for our local encoder. To introduce $\mathbb{SO}(3)$-equivariance to the features, we incorporate Vector Neurons [26] into our network. This approach, as described in the Vector Neurons paper, helps ensure that the features maintain equivariance with respect to rotations in three-dimensional space. Overall, the local encoder comprises approximately 1.7 million trainable parameters, allowing it to capture and encode the relevant local information from the point clouds.

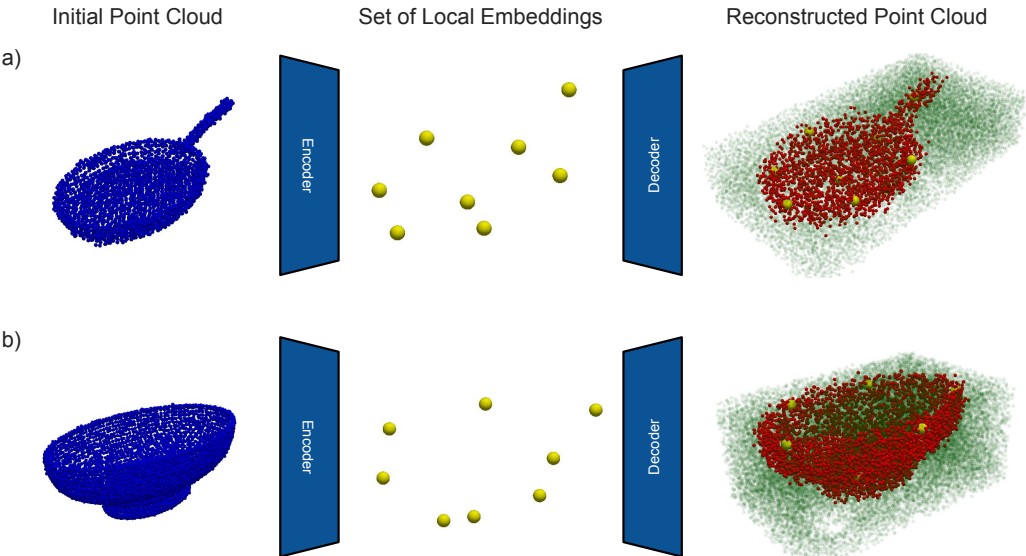

Figure 8: Examples of of our trained auto-encoder when reconstructing a pan (a), and a bowl (b). Blue point clouds represent initial point cloud observations, yellow points represent sampled centroids, and red and green points represent network prediction made for that point, occupied (red) and not occupied (green).

To enforce, that the local embeddings indeed encode the local geometry, we pre-train them as an implicit occupancy network [27], where a decoder is given a query point and a local feature embedding and is asked to determine whether a query point lies on the surface of the encoded part of the object $D_\theta(\mathcal{F}^i, q) \rightarrow [0, 1]$. Decoder is implemented as a PointNet Model [34] with GeLU activation functions [35] (without Vector Neurons).

We utilise positional encoding [28] and express edge features as $q = (sin(2^0 \pi p_q), cos(2^0 \pi p_q), ..., sin(2^{L-1} \pi p_q), cos(2^{L-1} \pi p_q))$, where $p_q$ is the position of the query point, and $L$ is the number of frequencies used. In our experiments, we set $L = 10$.

To train the occupancy network as an auto-encoder (as presented in Figure 8), a synthetic dataset is generated, consisting of point clouds for randomly sampled ShapeNet objects and corresponding labelled query points obtained using the PySDF library. This dataset comprises a total of 100,000 samples. During the training process, two NVIDIA RTX 2080ti GPUs were utilised for computational acceleration. The training duration spanned a period of approximately 3 days. We used AdamW [36] optimiser and scheduler our learning rate using the Cosine Annealing scheduler.

## C    Energy Based Model

To learn our proposed alignment distribution $p_\theta(\mathcal{A}^t_{test}|\mathcal{A}^t_{demo})$, we employ an energy-based approach and model the distributions as:

$$p_\theta(\mathcal{A}_{test}|\mathcal{A}_{demo}) = \frac{E_\theta(\mathcal{A}_{test}, \mathcal{A}_{demo})}{\mathcal{Z}(\mathcal{A}_{test}, \theta)} \tag{3}$$

Here $\mathcal{Z}$ is a normalising constant. In practice, we approximate this, otherwise intractable constant using counter-examples and minimise the negative log-likelihood of Equation 1

### C.1    Architecture

We are using heterogeneous graphs constructed using features described in Section B to represent the alignment between two objects $\mathcal{G}(\{\mathcal{F}^i_A, p^i_A\}^K_i, \{\mathcal{F}^i_B, p^i_B\}^K_i)$. Edges in the graph in the alignment are represented as relative positions between nodes expressed using positional encoding as:

$$e_{ij} = (sin(2^0\pi(p_j - p_i)), cos(2^0\pi(p_j - p_i)), ..., sin(2^{L-1}\pi(p_j - p_i)), cos(2^{L-1}\pi(p_j - p_i))) \tag{4}$$

In our base model, we use $L = 6$. Nodes in the demonstration and test alignment graphs are connected with **direction** edges equipped with learnable embeddings, effectively propagating information about the demonstration alignments to the test alignment graph. Note, that we are using heterogeneous graphs, meaning different edges (connecting nodes from the same object, target and grasped objects, and connecting demonstration and test graphs) have different types and will be processed with separate learnable parameters. Finally, to make predictions based on the connected graphs, we add an additional type of node to the graph, which aggregates the information from the test alignment graph. This Node can be seen as a $Class$ token, and each of the considered alignments in a batch (number of counter-examples + 1) is connected to a separate $Class$ node.

Having the designed graph structure, we use graph transformer convolutions, which can be viewed as a collection of cross-attention modules. These modules facilitate message passing between nodes in the graph, taking into account the specific types of nodes and edges in our heterogeneous graph representation. For a specific type of nodes and edges in the graph, the message passing and attention mechanism can be expressed as:

$$\mathcal{F}'_i = W_1\mathcal{F}_i + \sum_{j \in \mathcal{N}(i)} \alpha_{i,j}(W_2\mathcal{F}_j + W_6 e_{ij}); \quad \alpha_{i,j} = softmax\left(\frac{(W_3\mathcal{F}_i)^T(W_4\mathcal{F}_j + W_6 e_{ij})}{\sqrt{d}}\right) \tag{5}$$

Embedding from the $Energy$ node (or $Class$ token) is then processed with a small MLP to produce the predicted energy.

Our base model is comprised of 4 graph transformer convolutions with 4 multi-head attention heads, each with a dimension of 64. Final MLP is composed of 2 layers (with dimensions 256) and GeLU activation functions [35]. The full model contains around $5.7M$ trainable parameters.

### C.2    Training

To train our proposed energy model, we first need to create alignments of test objects used as counter-examples $\{\hat{\mathcal{A}}^j_{test}\}^{N_{neg}}_j$. We do so by creating copies of $\mathcal{G}_{test}(\{\mathcal{F}^i_A, p^i_A\}^K_i, \{\mathcal{F}^i_B, p^i_B\}^K_i)$, and applying $\mathbb{SE}(3)$ to the nodes in the graph describing the grasped object. Note that demonstration alignment graphs do not need to be copied, as they are connected to the test alignment graph with directional edges, propagating information one way.

To actually transform the nodes in the graph corresponding to the grasped object, both, the position and the feature vectors need to be transformed. Given a transformation $T_{noise}$, and it's corresponding rotation matrix $R_{noise}$ we update the graph nodes as:

$$[\hat{p}_A, 1]^T = T_{noise} \times [p_A, 1]^T \quad \hat{\mathcal{F}}_A = R_{noise} \times \mathcal{F}_A \qquad (6)$$

Note that this is possible because of the use of SE(3)-equivariant embeddings described in Section B. During training, we create 256 different $\hat{\mathcal{A}}_{test}$ alignments per batch to approximate $\mathcal{Z}$, each created using a unique $T_{noise}$. All the counter-examples as alternative graph alignments are created directly on a GPU, facilitating an efficient training phase.

To calculate a set of transformations $T_{noise}$ we use a combination of Langevin Dynamics (described in Section C.4) and uniform sampling at different scales. We start the training with uniform sampling in ranges of $[-0.8, 0.8]$ metres for translation and $[-\pi, \pi]$ radians for rotation. After $N$ number of optimisation steps ($10K$ in our implementation), we incorporate Langevin Dynamics sampling which we perform every 5 optimisation steps. During this phase, we also reduce the uniform sampling range to $[-0.1, 0.1]$ metres for translation and $[-\pi/4, \pi/4]$ radians for rotation. Although creating negative samples using only Langevin Dynamics is sufficient, in practice, we found that our described sampling strategy leads to faster convergence and more stable training for this specific application of energy-based models.

All models were trained on a single NVIDIA GeForce 3080Ti GPU for approximately 1 day.

During the training of the proposed energy model, several important tricks were employed to ensure stability, efficiency, and smoothness of the energy landscapes for effective gradient-based optimisation. These tricks contribute to the overall training process and facilitate the convergence of the model. The following tricks were identified as particularly significant:

- $L2$ **Regularisation**: To prevent the logits from diverging towards positive or negative infinity, a small $L2$ regularisation term is added to the loss function. This regularisation term helps to control the magnitude of the logits and maintain stability during training.

- **Spectral Normalisation**: Spectral normalisation is applied to all layers of the network. It normalises the weights of the network using the spectral norm (the largest singular value) of each weight matrix. In our case, energy landscapes that were learnt without using spectral norms were unusable for gradient-based optimisation.

- $L2$ **Gradient Penalties**: Gradient penalties are applied to the feature vectors of edges connecting grasped and target objects. This technique imposes an $L2$ regularisation on the gradients, penalising large changes in the input space. By doing so, the energy landscape becomes smoother and more amenable to gradient-based optimisation.

- **Pre-training on a Subset of the Data**: When dealing with a large and diverse dataset, it is beneficial to initialise the network by pre-training it on a smaller subset of the training data. This pre-training process allows the gradients to flow in regions of the loss-function landscape that would otherwise be relatively flat. As a result, the network can start from a better initialisation point, accelerating the training process. In the specific case mentioned, pre-training on approximately 1,000 samples saved approximately 70% of the total training time.

- **Dividing search space between multiple models**: We train two models that are exactly the same, but one is trained without rotations, while the other one is trained with rotations only. It reduces the search space each model needs to generalise over and prevents conflicting gradients during inference.

- **Mix negative sampling strategy**: It allows the model to initially learn coarse energy landscape (using random sampling), which is later refined using small perturbations and Langevin sampling.

- **Regular checkpoint evaluation**: It allows to detect training instabilities and select a stable checkpoint with a smooth energy landscape suitable for gradient-based optimisation.

Although training energy models using a contrastive learning approach can be challenging and unstable, the discussed training techniques, together with the use of a joint graph representation, that

mitigates the need for the network to learn a highly non-linear mapping between observation and action spaces, lead to stable and efficient training. Figure 9 shows typical training logs of our model including training loss and maximum and minimum predicted energies throughout training.

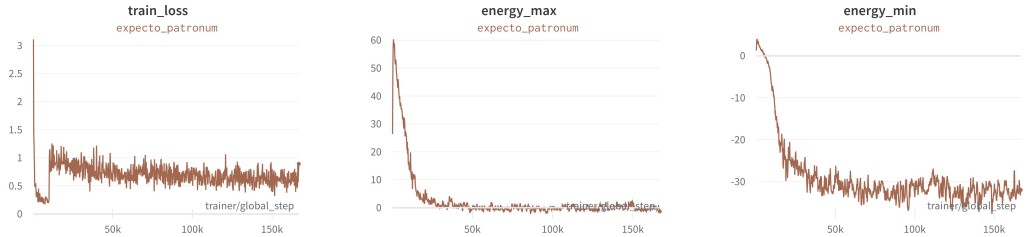

Figure 9: Training logs of our energy model (translation). Please note that the sudden jump in the training loss is due to the discussed change in the negative sampling strategy.

### C.3 Scaling to the Number of Demonstrations

As discussed previously, nodes in the graph that belong to the same demonstration are densely connected using different types of edges. Nodes from demonstrated and test alignments are connected with directional edges (there are no edge connections between demonstrated alignments themselves). Because of this, we can connect demonstrated alignments to any number of parallel test samples and make energy predictions for all of them in a single forward pass. Therefore, the number of nodes in the graph grows linearly with the number of demonstrations, but the number of edges also grows linearly. More precisely, the number of edges in the graph grows as $N \times M \times K$, where $N$ is the number of demonstrations, $M$ is the number of parallel samples, and $K$ is the number of nodes per demonstration. Figure 10 shows graphs depicting how the number of nodes, the number of edges, memory usage and time required for a single forward pass depends on the number of demonstrations used.

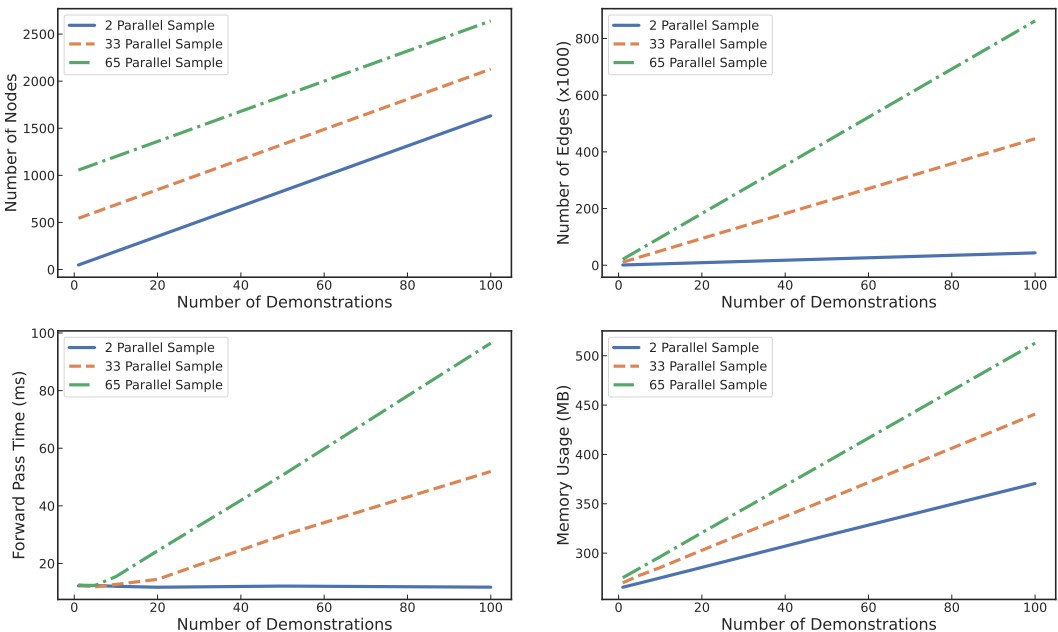

Figure 10: Graphs showing how various metrics of our heterogeneous graph representation scale with the number of demonstrations used.

## C.4  Inference Optimisation

Assuming a learnt, previously described, energy-based model, our goal at inference is to use it to sample from the conditional distribution $p_\theta(\mathcal{A}_{test}|\mathcal{A}_{demo})$. We can not directly sample alignments of objects $\mathcal{A}_{test}$, but we can compute $\mathbb{SE}(3)$ transformation $\mathcal{T}$, that when applied to the grasped object would result in an alignment between the objects that are within the distribution $p_\theta(.)$.

To solve Equation 2, we utilise an iterative gradient-based approach (Langevin Dynamics sampling). Each iteration step $k$ in the optimisation process updates the nodes of the graph alignment representation corresponding to the grasped object as:

$$[p_A^{k+1}, 1]^T = \frac{\lambda}{2} T_{update}^k \times T_{noise}(\epsilon^k) \times [p_A^k, 1]^T, \quad \mathcal{F}_A^{k+1} = R_{update}^k \times R_{noise}(\epsilon^k) \times \mathcal{F}_A^k \quad (7)$$

Here, $\epsilon^k \sim \mathcal{N}(0, \sigma_k^2) \in \mathbb{R}^6$ and $T_{noise}$ is calculated using exponential mapping to project it to $\mathbb{SE}(3)$ as $T_{noise}(\epsilon) = \text{Expmap}(\epsilon)$. In practice, to calculate $T_{update} \in \mathbb{SE}(3)$ (and $R_{update} \in \mathbb{SO}(3)$), we first transform the appropriate nodes in the graph using an identity transformation $T_I \in \mathbb{SE}(3)$ and calculate its gradient using back-propagation as $\nabla_{T_I} E_\theta(\mathcal{A}_{test}(T_I \times O_A^k, O_B), \mathcal{A}_{demo}) \in \mathbb{R}^6$. Finally, $T_{update}$ is calculated by taking an exponential mapping of $\nabla_{T_I} E_\theta(\cdot)$ as $T_{update}^k = \text{Expmap}(\nabla_{T_I} E_\theta(\cdot))$.

# D Experiments

## D.1 Task Definitions

We evaluate our approach on six different tasks: 1) *Grasping*. The goal is to grasp different pans by the handle, where success means the pan is lifted by the gripper. 2) *Stacking*. The goal is to stack two bowls, where success means one bowl remains inside the other bowl. 3) *Sweeping*. The goal is to sweep marbles into a dustpan with a brush, where success means that 2 out of the 3 marbles end up in the dustpan. 4) *Hanging*. The goal is to hang a cap onto a deformable stand, where success means the cap rests on the stand. 5) *Inserting*. The goal is to insert a bottle into a shoe, where success means the bottle stays upright in the shoe. 6) *Pouring*. The goal is to pour a marble into a mug, where success means the marble ends up in the mug.

## D.2 Implementation Details for Real-World Experiments

**Obtaining Point Clouds**: to obtain point cloud observation for our real-world experiments we are using 3 calibrated RealSense D415 depth cameras. To calibrate the extrinsics, we are using a standard charuco board and calibration implementation from the open-cv library. Typical calibration errors that we observe are in the range of $2 - 5$ mm. To combine observations from different cameras, we first projected separate point cloud observations to a common frame of reference (frame of the robot base) using calibrated extrinsics. We then concatenated the point clouds and used voxel downsampling (voxel size of 5 mm) to remove the redundant information. We first capture the point cloud of the workspace using the three cameras (with the gripper empty) to obtain the observation of the target object. We then place the object in the robot's gripper. The robot then moves the end-effector to the pre-defined position in which the grasped object is seen from 2 external cameras, and the observation of the grasped object is captured. If the task does not involve the grasped object (e.g. grasping), we are using the point cloud of the gripper obtained in simulation. Typical point clouds that we obtained using such a procedure can be seen in Figure 11.

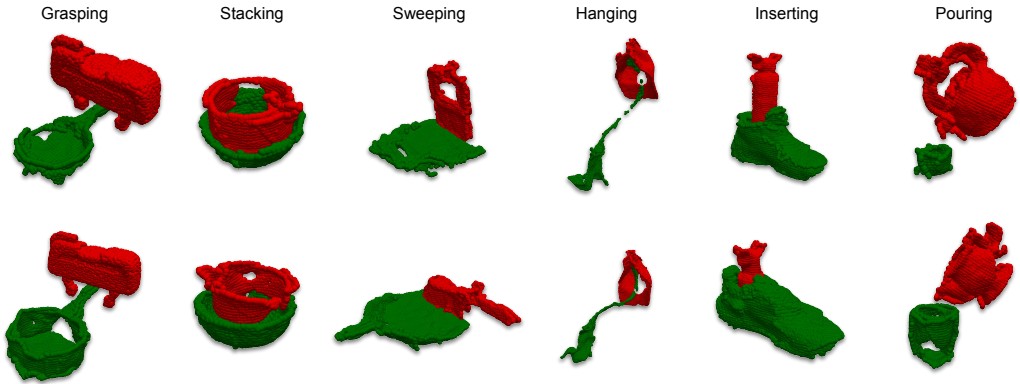

Figure 11:

**Recording Demonstrations**: We represent the demonstration trajectories as a series of waypoints of alignments between two objects. To obtain these waypoints of alignments of point clouds using kinesthetic teaching, we assume rigid grasps and utilise the forward kinematics of the robot. When the grasped object is moved to the desired waypoint, we record the pose of the end-effector and apply a transformation to the point cloud of the grasped object, such that it is transformed to the current pose. This mitigates the need to re-capture point cloud observations at each waypoint. Finally, all the point clouds recorded are transformed in the robot's end-effector frame. In this way, the predicted transformation can be directly applied to the robot's end-effector, reaching the predicted waypoint.

**Executing the Predicted Trajectory**: In this work, we model the trajectories as a series of sparse waypoints that need to be reached in sequence. Therefore, we use a standard position controller that interpolates the path between the waypoints and reaches them with constant velocity.

## D.3  Discussion on Failure Cases

The failure cases of our method were mainly observed when the objects used have significantly different geometry from the ones to provide demonstrations. This is evident from the performance of our method on the *Sweeping Task*, where the brushes used to provide demonstrations are significantly different from the ones used during testing (see Figure 12 top). The shape augmentation model we used to create the dataset of consistent alignments can not create such differences in augmented objects, making them significantly out of distribution for our trained model. Figure 12 bottom shows types of deformations that could be created from the demonstration objects. Note that such failure cases could be mitigated by using more powerful shape augmentation models coming from the fastly moving 3D vision community.

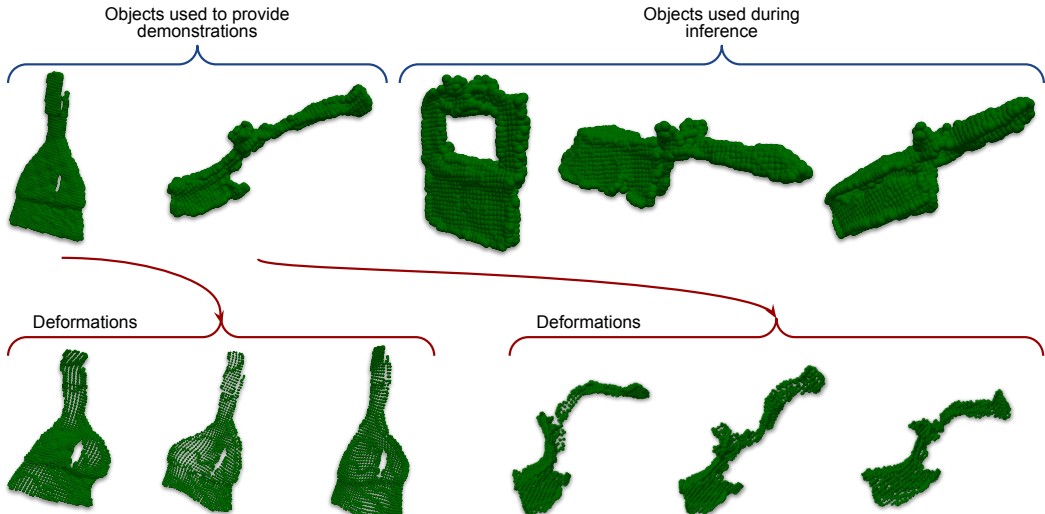

Figure 12: Top - point cloud observations of brushes used for the *Sweeping Task*, Bottom - types of shape deformations produced by DIF-Net [2] from the objects used for demonstrations.

## D.4  Exploring Demonstration Diversity

To explore the importance of diversity in the demonstrations, we created 3 different evaluation datasets, each with increasing diversity of objects. We vary the diversity of the objects by increasing the random scaling range, as well as the scale of the latent vector $\alpha$, which controls the amount of deformations applied by the DIF-Net model [2]. Examples of point cloud samples from the created datasets can be seen in Figure 13.

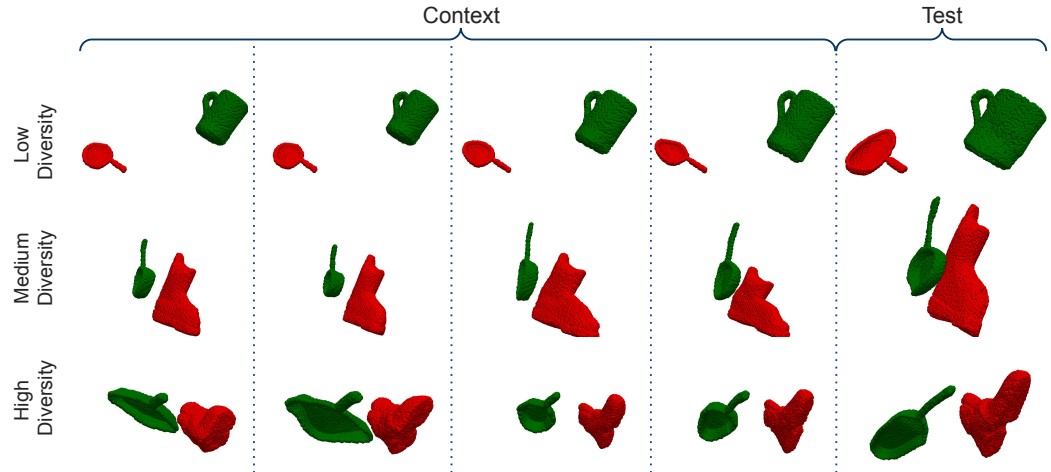

Figure 13: Random samples from the 3 different datasets used for the data diversity experiment. Green point cloud represent object $A$, while red point cloud represent object $B$.

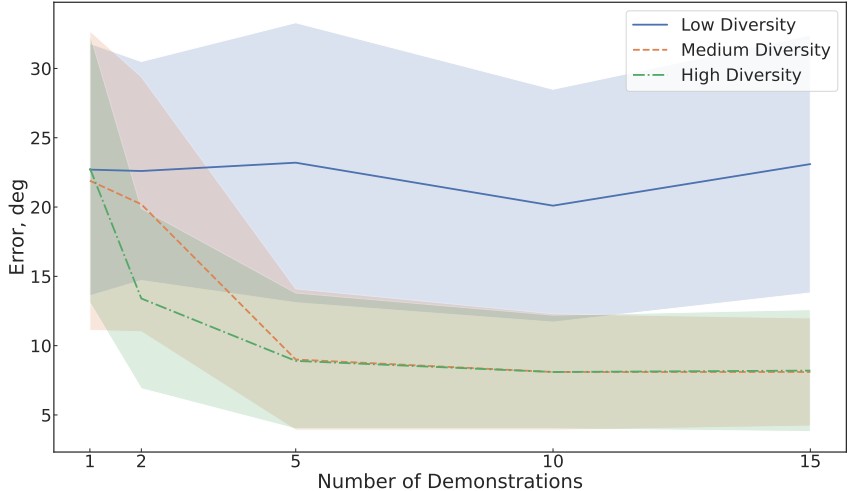

Figure 14: Rotational error based on the number of demonstrations for 3 different sets of diversities.

