# OpenReview forum: "Few-Shot In-Context Imitation Learning via Implicit Graph Alignment"
_robot-learning.org/CoRL/2023/Conference — CoRL 2023 Poster_

### Official Review · Reviewer_4Z1U · 2023-07-16

**Confidence:** 4
**Originality:** Very Good
**Technical Quality:** Excellent
**Clarity Of Presentation:** Very Good
**Impact:** 4

**Recommendation:**

Strong Accept: I recommend accepting the paper and will argue for my recommendation even if other reviewers hold a different opinion.

**Review:**

Strengths:
- The work demonstrates few shot imitation learning of object pair alignment which they show generalizes to novel, unseen objects and clear evaluations describing performance gains upon prior work
- The explanation for the task and training / deployment time pipelines are well written
- The method is clear in how it approaches novel object / pose generalization, with the augmentation based pipeline for synthesizing training data and energy model which does not make assumptions about specific object categories / types
- The experiment procedures in sim/real are described well and are convincing in showing generalization capabilities (especially beyond prior works), the few-shot / sample requirements and tradeoffs, and real world performance

Weaknesses:
- Though the work claims that it can be used to solve several / generic robotic tasks, all tasks are relatively similar in being a correct alignment problem between an object grasping in a robotic gripper and a static object. Specifically, some of the tasks feel artificial for this setting, for example in the sweeping task, it appears that the marbles are directly placed within the path between the dustpan and brush, rather than using perception / reasoning to clean up all marbles.
- Figure 4 and Appendix Figure 1 are appreciated, though a more detailed figure presenting the global view of the proposed framework / system would make the method more clear, given the broad set of works leveraged / described in Sections 3.2 to 3.4, e.g. DIF-Net, Further Point Sampling, Set Abstraction, Vector Neurons, the occupancy network, positional encodings, and the transformer energy model.

Misc:
- On line 87, missing word: “defining this analytically is infeasible [for] all possible task and …”


**Quality Of The Limitations Section:**

Limitations are addressed clearly

**Questions For Rebuttal:**

- Were experiments run with multiple random seeds for training?
- Could confidence intervals for the real-world experiments also be provided, to show statistical significance?


**Robotics Focus:**

Sufficient demonstration on hardware

**Summary Of Paper:**

The work presents a new method for few-shot imitation learning of object manipulation tasks via aligning object-level graph representations. Specifically, the system focuses on one-robot arm manipulation of one object in relation to another object using <15 demonstrations and 3 depth cameras for inputs. The main novelty of this work, compared to prior art in R-NDF and TAX-Pose, is using an energy-based model / graph which can better generalize to unseen object categories and alignments compared to the former occupancy networks. Generalization is further achieved using correspondence-preserving shape augmentation (procedurally). Experiments in simulation show that the model generalizes better across seen/unseen alignments and objects. Experiments in real show show outperformance against baselines.


**Summary Of Recommendation:**

Overall, the work is convincing in demonstrating few-shot imitation learning for a broad set of objects and tasks which is well supported by experiments, ablations, and the method. Given the visuals and videos, it is clear that the results demonstrated have promise on real hardware and are relevant for addressing potentially important robotics problems (grapsing, pouring, stacking, sweeping), especially given the formulation designed around invariance to object domain and specific detail.

---

### Official Review · Reviewer_r1Xw · 2023-07-17

**Confidence:** 4
**Originality:** Good
**Technical Quality:** Very Good
**Clarity Of Presentation:** Very Good
**Impact:** 2

**Recommendation:**

Strong Accept: I recommend accepting the paper and will argue for my recommendation even if other reviewers hold a different opinion.

**Review:**

Strengths:
1. The main idea of the paper, i.e. modeling the relative alignment of objects conditioned on the task using a graph-based transformer energy model, is interesting. While certain assumptions are made, as detailed in the Limitations section, the work serves as a proof-of-concept for the approach, which is worth exploring further.
2. The experiments are thorough, and look at various research questions. Specifically, the authors experiment with 5 scenarios for generalization: seen alignments, unseen alignments, unseen object instances, unseen object categories, multimodal demos. The results show that the proposed approach is effective across all scenarios. The second set of experiments shows that diverse demos are required to improve generalization. Finally, real-world experiments on 6 tasks shows that the proposed approach is effective.
3. The paper is well-written and is easy to follow.

Weaknesses:
1. The title of the paper could be a little misleading, given that the term "in-context learning" is used primarily for LLMs these days.

**Quality Of The Limitations Section:**

Limitations are addressed clearly

**Questions For Rebuttal:**

1. Is the same conditional alignment distribution used for both simulated and real-world experiments? If not, how is the training data for real-world tasks generated?

**Robotics Focus:**

Sufficient demonstration on hardware

**Summary Of Paper:**

This paper looks at the problem of generalizing robotics tasks to new objects in an imitation learning setting. The problem is formulated as inferring a trajectory of alignments between a grasped object and a target object. The proposed approach involves learning a task-conditioned alignment distribution, which is modeled as a graph-based transformer energy model. The dataset for training is created using ShapeNet.
Experimental evaluation looks at generalization with novel objects and novel object categories, the effect of diversity in demonstration, and real-world experiments.

**Summary Of Recommendation:**

The paper introduces a novel method for generalizing to new objects. The presented idea has been effectively validated using several experiments, under certain assumptions (as listed in the Limitations section). Overall, this is a good proof-of-concept, and the research direction can be further explored by the community.

---

### Official Review · Reviewer_UYjn · 2023-07-20

**Confidence:** 2
**Originality:** Good
**Technical Quality:** Good
**Clarity Of Presentation:** Fair
**Impact:** 4

**Recommendation:**

Weak Accept: I recommend accepting the paper, but will not argue for my recommendation if the majority of other reviewers have a different opinion.

**Review:**

The paper introduces a unique approach aimed at achieving demonstrated alignments between two objects. The problem is interesting and the proposed method is novel. However, it lacks a comprehensive description of its implementation, with several key details missing (see questions below)

**Quality Of The Limitations Section:**

Limitations are addressed clearly

**Questions For Rebuttal:**

- In the limitations section, the authors claim that the model can effectively handle noisy observations. However, the extent of this noise tolerance remains unclear. The level of noise the method can accommodate needs further explanation.
- The implementation of the script controller for real-world experiments is not clearly detailed.
- How are point clouds obtained from the three cameras in these real-world scenarios?
- It would be helpful if the authors could provide visualizations highlighting the types of unseen objects the model fails to handle effectively. A comparison detailing how these unseen objects differ from the ones that the model has seen could provide valuable insights.

**Robotics Focus:**

Sufficient demonstration on hardware

**Summary Of Paper:**

This paper introduces a solution for imitation learning that takes a minimal number of demonstrations. The idea is learn graph representations to enable the model to conduct in-context learning when executing tasks with a new set of objects, thereby eliminating the need for additional training.

The method takes point cloud inputs and employs ShapeNet as its training data source. An object augmentation function is utilized for data augmentation. To construct the dataset, two random objects are chosen and deformed using a generative model, while their relative poses remain unchanged. The graph representation is used to get an energy score between multiple objects pairs. Then a series of transformations was learned to minimize this score.

Both simulated and real-world experiments were conducted. Tasks in real-world experiments included grasping, stacking, sweeping, hanging, pouring, and inserting.

**Summary Of Recommendation:**

The proposed method appears to have a strong potential in generalizing to novel objects and scenes, a capability that is crucial in many real-world applications. While the methodology could benefit from additional clarity in certain aspects such as noise tolerance levels and the script controller implementation for real-world experiments, it already provides a solid foundation. However, it should be noted that my assessment is made without a deep expertise in this direction.

I've read the rebuttal and my concerns have been addressed. The additional result on failure cases looks interesting and convincing.

---

### Official Review · Reviewer_R4fh · 2023-07-21

**Confidence:** 4
**Originality:** Very Good
**Technical Quality:** Very Good
**Clarity Of Presentation:** Very Good
**Impact:** 4

**Recommendation:**

Weak Accept: I recommend accepting the paper, but will not argue for my recommendation if the majority of other reviewers have a different opinion.

**Review:**

Strengths:
- The idea and algorithm of learning an implicit alignment by optimizing an energy function based on a graph built between the test-time objects and demonstration objects is quite novel.
- The experiments evaluations are well done in both simulation and in the real-world, and the method is shown to perform better than prior methods, with better generalization abilities.
- The paper is clearly written and well-motivated.

Weakness:
- See Questions for rebuttal.

**Quality Of The Limitations Section:**

Limitations are addressed clearly

**Questions For Rebuttal:**

I have 2 major questions:

- It is well known that training with Info-NCE loss to optimize the energy function can be instable. Did the authors encounter similar issues with this, and if so, how is this solved?
- How can the method scale up with the number of demonstrations? It seems that the number of graph nodes would grow linearly with the number of demonstrations, and the number of edges would grow quadratically (the paper did not mention how the edges are constructed for the graph, so a clarification would be needed). This seems to make the method impractical for a large number of demonstrations.

**Robotics Focus:**

Sufficient demonstration on hardware

**Summary Of Paper:**

This paper proposes a new method for few-shot imitation learning named implicit graph alignment, which generates an alignment for two test-time objects conditioned on a few demonstrations. The alignment, which is a SE3 transformation, is generated by optimizing a learned energy function, whose input is a heterogenous graph built between the test-time and demonstration objects. The energy function is trained in simulation using shape augmentation via a InfoNCE loss such that test-time alignments that matches the demonstration alignments have low energy. The method is evaluated in simulation and on 6 real-world tasks and achieves lower alignment error compared to previous methods, and shows good generalization towards novel object instances/categoriess and spatial configurations.

**Summary Of Recommendation:**

See as above. I am open to increase the score if my concerns are addressed during the rebuttal period.

---

### Author Response · Authors · 2023-08-11
**General Response**

Thank you very much to all reviewers for your insightful questions. Since there were no common issues or questions amongst reviewers, we have written our responses individually to each reviewer. We have also updated our paper based on your feedback, with the following updates (all links are anonymised):

* Created new visualisations of the gradient-based optimisation used at inference (see our [anonymised webpage](https://sites.google.com/view/implicit-graph-alignment)).
* Expanded the discussion on stability during training in the supplementary material ([see typical training curves here](https://drive.google.com/file/d/1niKhOcEKgtIcNY0hXtffhOIO4wLBU9Gk/view?usp=sharing)).
* Created a visualisation of how the learnt energy landscape evolves during training ([link](https://drive.google.com/file/d/1DYJpcPbno2rXEzGPS07oxf50lTQYsGWP/view?usp=sharing)).
* Added discussion and graphs showing how time and memory requirements of our method scale with the number of demonstrations ([link](https://drive.google.com/file/d/1arn11Y06Pwd0T0m0zcqZU6JYTSq5sYkE/view?usp=sharing)).
* Updated Figure 4, to more clearly illustrate the structure of the graph representation ([link](https://drive.google.com/file/d/1PZ6cGgnXSmhq5OZsrhePX3nDIl0uAzTI/view?usp=sharing)).
* Added description of the implementation details for the real-world experiments to the supplementary material, together with the visualisation showing examples of point clouds obtained in the real-world ([link](https://drive.google.com/file/d/1IcpzMrclBIX89m6CUCSkD5z6T1xOCekw/view?usp=sharing)).
* Added discussion on the most common failure cases to the supplementary material, together with the figure showcasing the main cause (significant variations between demonstration and test objects) ([link](https://drive.google.com/file/d/1aVPQaJ4xD4QO5PhbkugstAz0yY3hkKJ9/view?usp=sharing)).
* Added a figure to the supplementary material illustrating the global overview of the proposed framework ([link](https://drive.google.com/file/d/1JmdVABMUjOvAiKrDBQ27YAUyhezQmt_Q/view?usp=sharing)).
* Added aggregate performance metrics across all real-world tasks to the results table.

We have uploaded our updated paper and supplementary material alongside a reply to each reviewer individually, with all new material in blue text. And to see our new visualisations and animations, please visit our updated anonymised webpage at [https://sites.google.com/view/implicit-graph-alignment](https://sites.google.com/view/implicit-graph-alignment).

There are still a few days remaining in the discussion period, so if reviewers have any further questions, then please do ask and we would be very happy to discuss.

---

### Decision · Program_Chairs · 2023-08-30

**Decision:**

Accept (Poster)

**Comment:**

This paper introduced an algorithm that can make imitation learning policies generalize to new objects at test time. The key idea is to learn an implicit alignment between the test-time objects and the demonstration objects by optimizing an implicit energy function. The experiments demonstrated the effective few-shot learning capabilities of this method on real-world manipulation tasks. This paper received positive ratings from all four reviewers. The reviewers appreciated the novelty of the proposed approach and the real-world experimental validations. Some clarification questions have been raised and properly addressed by the rebuttal.

However, one lingering issue is the choice of real-world tasks. As 4Z1U correctly pointed out, "all tasks are relatively similar," and "some of the tasks feel artificial." It remains to be seen whether this approach could scale up to handle the increased complexity of real-world, everyday tasks. Nonetheless, considering the contributions made in this paper and the unanimous support from the reviewers, the AC believes that this paper has passed the bar of acceptance at CoRL and recommends accepting it for this conference.